# Video-Urodynamic Characteristics and Predictors of Switching from Botulinum Neurotoxin a Injection to Augmentation Enterocystoplasty in Spinal Cord Injury Patients

**DOI:** 10.3390/toxins14010047

**Published:** 2022-01-10

**Authors:** Chih-Chieh Lin, Hann-Chorng Kuo

**Affiliations:** 1Department of Urology, Taipei Veterans General Hospital, Taipei 112304, Taiwan; jayslylin@gmail.com; 2Department of Urology, School of Medicine, College of Medicine, Shu-Tien Urological Research Center, National Yang Ming Chiao Tung University, Taipei 112304, Taiwan; 3Department of Urology, Hualien Tzu Chi Hospital, Buddhist Tzu Chi Medical Foundation, Tzu Chi University, Hualien 970374, Taiwan

**Keywords:** augmentation enterocystoplasty, botulinum neurotoxin A, patients, spinal cord injury, urodynamic parameters, urodynamic study

## Abstract

Botulinum neurotoxin type A (BoNT-A) injection and augmentation enterocystoplasty (AE) are alternative and effective management strategies for neurogenic detrusor overactivity (NDO) refractory to pharmacotherapy. A great majority of patients with spinal cord injury (SCI) may, however, prefer BoNT-A injections to AE, due to the less invasive characteristics. In this study we evaluated the influence of various video-urodynamic study (VUDS) parameters in SCI patients who continuously received repeat BoNT-A detrusor injections or switched to AE to improve their bladder conditions. We compared the changes in the urodynamic parameters before and after each mode of treatment. In this retrospective study, all SCI patients with refractory NDO who had received at least one BoNT-A injection were enrolled. VUDS was performed before and after both BoNT-A injection and AE. All of the urodynamic parameters of the storage and micturition—including the bladder capacity of every sensation, maximal flow rate (Qmax), post-voiding residual volume, detrusor pressure at Qmax, and bladder contractility index—were recorded. A total of 126 patients, including 46 women and 80 men, with a mean age of 41.8 ± 13.1 years, were recruited for this study. All of the patients receiving either BoNT-A injection or AE had a statistically significant increase of bladder capacity at every time-point during filling and a decrease in detrusor pressure at Qmax during voiding. Patients who switched from BoNT-A to AE had greater improvements in their urodynamic parameters when compared with those who continued with BoNT-A injections. Accordingly, SCI patients receiving BoNT-A injections but experiencing few improvements in their urodynamic parameters should consider switching to AE to achieve a better storage function and bladder capacity.

## 1. Introduction

Neurogenic detrusor overactivity (NDO) can be a bothersome neurogenic lower urinary tract dysfunction (NLUTD) in patients with suprasacral spinal cord lesions [1,2]. In addition, detrusor sphincter dyssynergia (DSD) may concurrently result in dysfunction of both the urinary bladder and external urinary sphincter during bladder storage and micturition [2,3]. These patients may suffer from lower bladder capacity as a result of potential structural damage to the bladder wall or high intravesical pressure [4]. Urodynamic studies usually present with detrusor overactivity (DO) with or without striated urethral sphincter dyssynergia (DSD), low bladder compliance, abnormal bladder sensations, urinary incontinence, and sometimes an incompetent or relaxing urethra [1,2,3,4,5]. Consequently, treatment of urinary problems in patients with DSD are intended to decrease the intravesical pressure, increase the bladder capacity, and control urinary incontinence [1,3,4,5].

According to the 2021 American Urological Association/Society of Urodynamics (AUA/SUFU) guidelines on adult neurogenic lower urinary tract dysfunction, pharmacological therapies (e.g., oral antimuscarinics) are considered a second line of treatment with respect to behavioral treatment [4]. Mainstream treatment usually includes antimuscarinic medications, timed voiding schedules, and clean intermittent catheterization (CIC) [5]. For refractory NDO, however, injection of botulinum neurotoxin type A (BoNT-A) into the detrusor muscle and augmentation enterocystoplasty (AE) have become alternative, but more invasive, treatment processes [1,3].

BoNT-A injection into the detrusor muscle was first approved by the United States Food and Drug Administration (FDA) in 2011 for the treatment of NDO [6]. BoNT-A produces a long-lasting (>6 months) but reversible chemical denervation [7,8,9,10]. Cruz et al. performed a randomized, placebo-controlled phase 3 study to demonstrate the effects of improving urgency incontinence episodes, increasing maximum bladder capacity, decreasing maximum detrusor pressure, and improving the quality of life (QoL) by intradetrusor injections of 200 U BoNT-A [11]. Thereafter in one three-arm (200 U, 300 U, and Placebo), phase 3 study, Ginsberg et al. revealed that the patient group with 200 U BoNT-A injections had a more favorable safety profile [12]. These studies revealed that BoNT-A intradetrusor injection can lead to significant improvements in urgency urinary incontinence, urodynamic parameters, and the ability to store urine at low pressures with fewer adverse effects [11,12,13]. The most commonly reported adverse events were urinary tract infections (UTIs) and elevated post-void residual urine (PVR) [11,12].

AE is regarded as a definite solution for urinary bladder with DO and poor compliance refractory to anticholinergic medications and BoNT-A injections. AE creates a large capacity reservoir and also decreases intravesical pressures and detrusor spasticity [14]. The goals of this treatment for spinal cord injury (SCI) patients are to increase bladder capacity and subsequently prevent upper urinary tract deterioration, but such patients have to empty the urinary bladder by CIC or spontaneous voiding [5]. Both AE and BoNT-A injection appear to improve bladder function and minimize urgency urinary incontinence [15].

Despite the published evidence for the effectiveness of BoNT-A injection and AE, few investigations through video-urodynamic study (VUDS) before and after treatment have focused on the differences between BoNT-A injection and AE [5,13,14,16,17]. Based on pre- and post-operative videourodynamic investigation, the aim of the present study was to assess if SCI patients submitted to BoNT-A injection achieve better functional outcomes from BoNT-A re-injection or from augmentation enterocystoplasty.

## 2. Results

In this study, a cohort of 128 SCI patients was recruited. Clinical (lesional level) and demographic characteristics (genders) of the patients are reported in Table 1. The mean age was 40.3 ± 12.9 years (range: 14–75 years) and the mean interval of VUDS before and after BoNT-A injection was 3.5 ± 3.5 years. Among the patients, 22 received VUDS evaluation after only one BoNT-A injection, 31 after two injections, 19 after three injections, 12 after four injections, 10 after five injections, 15 after six injections, 3 after seven injections, 2 after eight injections, 2 after nine injections, 5 after 10 injections, 5 after 11 injections, and 2 after 12 injections.

Table 2 compares the VUDS parameters of the 128 patients before and after BoNT-A injections. All patients had statistically significant improvements in sensation at different time points during the filling phase after BoNT-A injection. The VUDS parameters of the bladder sensation, including the first sensation of bladder filling (FSF), first desire to void (FS), and urgency sensation (US), all revealed a statistically significant increase in the bladder capacity at each time point (*p* < 0.01). The cystometric bladder capacity (CBC) increased significantly from 212 ± 119 to 359 ± 186 mL (*p* < 0.001). The maximal flow rate (Qmax), the detrusor pressure at Qmax (Pdet.Qmax), and the voiding efficiency (VE) all experienced statistically significant decreases after BoNT-A injection (*p* < 0.05). The PVR after BoNT-A injection was elevated significantly from 156 ± 114 to 303 ± 198 mL (*p* < 0.001). The mean bladder contractility index (BCI) in male patients underwent a significant decrease, from 55.8 ± 39.1 to 38.9 ± 31.9, after BoNT-A injection.

In these SCI patients with BoNT-A injection, 25 (19.5%) eventually received AE because of the observation of decreased bladder capacity and increased detrusor pressure in VUDS after the BoNT-A injections. Table 3 reveals the VUDS parameters of the patients without AE at times before and 3 to 6 months after BoNT-A injection. All the VUDS parameters of sensation showed significant improvement in bladder volume after BoNT-A injection in the short-term database. Being similar to the parameters in Table 2, all the parameters regarding the micturition showed a significant decrease in the contractility and detrusor pressure (*p* < 0.001). Table 4 presents the VUDS parameters of the 25 patients who received AE, measured before BoNT-A injection, after BoNT-A injection and before AE, and after AE. We also investigated the differences in the VUDS parameters from these 25 patients between the time before BoNT-A injection and the time before AE. Interestingly, all the urodynamic parameters, including the bladder volume of sensation at each time point and micturition, showed no significant difference between before BoNT-A injection and before AE (see Table 5). The mean follow-up period of VUDS after AE was 3.62 ± 3.04 years (range: 0.10–8.75 years). After AE, the bladder volume at each time point of the filling phase underwent a statistically significant increase relative to that before AE. The mean CBC increased significantly from 196 ± 123 mL before AE to 484 ± 126 mL after AE. 

The mean Qmax decreased significantly from 3.7 ± 5.2 to 2.1 ± 4.8 mL/s after operation (*p* = 0.048). Moreover, the mean PVR after AE underwent a significant increase, from 144 ± 118 to 459 ± 147 mL (*p* < 0.001). The values of mean Pdet.Qmax and mean BCI after AE decreased significantly, to 9.6 ± 11.4 cmH_2_O (*p* = 0.005) and 20.6 (*p* = 0.004), respectively. The VUDS parameters of the patients after AE were also significantly better than those of the patients who received only BoNT-A injections.

Table 5 represents the changes in urodynamic parameters before and after BoNT-A injection or AE. In Table 4, the changed VUDS values before and after AE showed no significant difference compared to those of before and after BoNT-A injection in most UDS parameters, such as FSF, FS, Pdet.Qmax, Qmax, VE, and BCI. The increase of CBC and PVR after AE treatment was significantly larger than that after BoNT-A injection (*p* = 0.004 and 0.001, respectively).

Table 6 represents the rate of the videourodynamic characteristics and the changes of urodynamic parameters after BoNT-A injections between patients with chronic spinal cord injury who underwent AE and those continuing BoNT-A injections. The SCI patients switching from BoNT-A to AE had significantly more contracted and trabeculated bladder and vesicoureteral reflux during the filling phase than those patients remaining with BoNT-A under VUDS (*p* < 0.0001). In addition, the rate of tight external sphincter during voiding was also significantly higher in the SCI patients switching to AE (*p* = 0.002). Moreover, the changes of bladder storage sensation, including FS, US, CBC, and PVR, were significantly lower in the patients who received AE after the initial BoNT-A injection. The changes of Pdet and BCI were also significantly lower than for those patients who continued BoNT-A injections.

## 3. Discussion

This study reveals that SCI patients who continued to display suboptimal therapeutic effects after repeated BoNT-A detrusor injections were likely to receive AE. The VUDS parameters after AE improved significantly when compared with those before AE and were also better than those after repeat BoNT-A injections. Limited clinical improvements after BoNT-A injections predict the potential for performing AE in SCI patients.

In many meta-analyses, randomized control trials, and treatment guidelines, BoNT-A injection in the urinary bladder is considered as an effective therapy in patients with NLUTD due to multiple sclerosis or SCI [11,12,17,18]. When treating NDO patients in a randomized, placebo-controlled phase 3 study, Cruz et al. found that intradetrusor injections of 200 U BoNT-A had the effects of improving urgency incontinence episodes, increasing maximum bladder capacity, decreasing maximum detrusor pressure, and improving the quality of life (QoL) [11]. Another randomized, double-blind trial proposed by Herschorn et al. demonstrated that 300 U BoNT-A injections in the urinary bladder can improve the mean daily frequency of urinary incontinence episodes at week 6 after injection [19]. Nevertheless, the effectiveness of BoNT-A injection may last for only nine months; repeated detrusor injections of BoNT-A were necessary to maintain the therapeutic efficacy [19,20,21].

In a preliminary report of a previous study, we stated that repeated 200 U BoNT-A detrusor injections at six-month intervals provided satisfactory outcomes for patients with NDO and urinary incontinence; nevertheless, the patients’ adherence rate was not as high as we had expected, due to the lack of consistent efficacy in improving incontinence severity [22]. This study featured 84% patients who had received repeated BoNT-A injections as a result of recurrence of urinary incontinence symptoms. The urodynamic parameters of sensation, including the capacity of first sensation, urgency sensation, and CBC, were all improved significantly after the BoNT-A injections. Meanwhile, the voiding parameters underwent statistically significant decreases, implying a lower risk of damage to the upper urinary tract after BoNT-A injections. Koschorke et al. proposed that videourodynamic studies might be helpful for monitoring the effect of injections on bladder pressure; they recommended urodynamic investigations to routinely check for a bladder pressure of no more than 40 cm H_2_O in the storage phase [17].

Consequently, in this study we used urodynamic studies to investigate the effectiveness of repeated BoNT-A injections and the timing of switching to AE. Myers et al. compared CIC, CIC with BoNT-A injections, and CIC with AE by applying a questionnaire of neurogenic bladder symptoms score (NBSS) [15]. They concluded that CIC with AE led to better urinary function and greater satisfaction with urinary symptoms relative to the other two treatments. Anquetil et al. performed another prospective cross-sectional study to compare the QoL of patients with SCI receiving BoNT-A injections or AE [16]. They reached a similar conclusion of better QoL in the AE group, with the possible reasons being the long-time interval between the two sessions of BoNT-A injections and the persistent effectiveness of AE. In our present study, all of the patients received at least one BoNT injection, with only 19 patients (15%) undergoing AE thereafter. The most significant differences between the patients receiving AE and not receiving AE were decreasing bladder capacity in the storage phase and increasing detrusor contractility in the patients receiving AE. Patients with BoNT-A tended to receive AE when the urodynamic parameters reverted to the baseline state prior to BoNT-A injection. Thus, the clinical outcome of BoNT-A injection became less effective than the initial BoNT-A treatment. In 2003, Quek et al. reported a long-term follow-up urodynamic study of 26 patients with NLUTD who underwent AE; they confirmed the urodynamic benefits of increasing bladder capacity and decreasing storage pressures [14]. Their findings further imply that switching from BoNT injection to AE should be considered for those patients who have limited increase of bladder capacity and improvement of bladder compliance after BoNT-A injections.

In a systemic review, Hoen et al. demonstrated that AE can be effective for improving QoL, urodynamic parameters, and continence and for maintaining stable renal function [23]. In our present study, all of the patients who received AE displayed statistically significant improvements in bladder capacity at first desire and at urgency sensation as well as CBC after AE. Moreover, the values of Pdet.Qmax and BCI underwent statistically significant decreases after AE. In a previous report, we found that a total of 79 patients receiving AE displayed improvements in mean CBC from 145 ± 88.8 to 473 ± 126 mL (*p* < 0.0001) and mean bladder compliance from 8.21 ± 5.96 to 58.1 ± 51.1 mL/cmH_2_O (*p* < 0.0001) [24]. In the present study, we further compared the characteristics of patients who switched to AE after BoNT-A injections with those of patients who continued with BoNT-A injections. All of the urodynamic parameters of sensation during filling for the patients after AE were better than those for the patients who continued with the BoNT-A injections. Nevertheless, the patients after AE had significantly decreased Pdet.Qmax, decreased VE, and increased PVR compared with those who did not undergo AE. When compared with BoNT injections, however, long-term complications regarding post-operative bowel dysfunction have been reported only in patients having AE [25]. In our previous study, we found that persistent hydronephrosis, UTI, and chronic diarrhea were minor complications [24]. There were 41.8% of patients requiring medical treatment for recurrent UTI and 21.5% patients with post-operative chronic diarrhea reported in our previous data. From the data of the present study, the larger PVR and lower VE and BCI could partly account for the higher incidence rate of recurrent UTIs.

VUDS has been a routine investigation for patients with chronic SCI and NLUTD. One single VUDS test can investigate not the bladder function (such as bladder sensation, bladder compliance, and contractility), bladder outlet condition (BND, DSD, or urethral stricture), bladder morphology (trabeculation, diverticulum), and presence of vesicoureteral reflux or not. In the ICI recommendation of urodynamic testing, VUDS is recommended to be an essential investigation for NLUTD, especially when invasive procedure is to be undertaken. Table 5 shows the morphological changes between patients receiving BoNT-A—BoNT-A and BoNT-A—AE. Significantly more patients with contracted bladder, low bladder compliance, presence of vesicoureteral reflux, and a tight external sphincter during voiding would switch from BoNT-A to AE. These morphological changes during VUDs can help identify which SCI patients might need AE for their bladder management.

The principal limitation of this study is that it is a retrospective study with only a small number of patients receiving AE after BoNT-A injections. There were a few patients who did not undergo the videourodynamic study for evaluation of their treatment outcomes, due to time factors or patient factors. Another limitation is that we did not measure the quality of life index after BoNT-A and AE treatment. Moreover, the quality of life indexes were not routinely measured in all SCI patients. Lack of quality of life measurement is surely a weakness of this study and reduces the value of this study. Nevertheless, our data provide important information regarding the use of urodynamic tools to ensure better timing of the switching from BoNT-A injection to AE. Another limitation is that CIC data were not further analyzed in our database, for example, the frequency of CIC in a day and the duration of performing CIC. Notwithstanding this, the frequency of CIC would probably be adjusted in terms of increasing bladder capacity after BoNT-A injection or AE. It would be difficult to control for this interfering factor without performing prospective research.

## 4. Conclusions

Among patients with SCI receiving BoNT-A injections, those who undergo repeated BoNT-A injections should consider switching to AE if their urodynamic parameters revert to the status prior to the initial BoNT-A injection. Our data confirmed that both BoNT-A injections and AE provide urodynamic benefits, including increased bladder capacity and decreased storage pressure during the follow-up. For urological clinicians, our results provide information regarding the comparison of the patients who switch to AE and those who continue with BoNT-A injections and reveal that performing AE leads to greater improvements in the urodynamics parameters of the storage phase than maintaining BoNT-A injections alone. From the finding of VUDS imagery, those SCI patients switching to AE had a significantly higher rate of contracted and trabeculated bladder and vesicoureteral reflux during the filling phase than those who continued BoNT-A injections. Therefore, SCI patients receiving BoNT-A injections but experiencing little improvement in their urodynamic parameters should consider switching to AE to achieve a better quality of life. Hopefully, in the near future, prospective studies related to predicting the timing of switching BoNT-A injections to augmentation enterocystoplasty will take place, in order to depict a more complete picture of the disease entity. 

## 5. Materials and Methods

In this retrospective single-center study, all SCI patients with refractory neurogenic DO with or without DSD, requiring CIC or not, were enrolled. Inclusion criteria were being aged more than 18 years old and receiving BoNT-A injection in the urinary bladder. VUDS was routinely performed before and after BoNT-A injection and applied to exclude the diagnosis of detrusor underactivity, anatomical obstruction, or intrinsic sphincter deficiency. Those patients who had genitourinary malignancy, urinary tract infection at the time of recruitment, or other chronic systemic diseases (e.g., chronic renal impairment or congestive heart failure) were also excluded.

VUDS was performed with reference to the recommendations and standardizations of the International Continence Society [26]. All patients were submitted to VUDS with external sphincter elctromyography before and after Bo-NTA injections. The following VUDS parameters were assessed: bladder compliance, maximum cystometric capacity, detrusor overactivity (maximum detrusorial pressure during involuntary detrusor contractions), morphologic characteristics of the bladder, evidence or not of vesicoureteral reflux, morphologic aspects of the bladder and of the urethra during leakage, and bladder residual volume. Pre- and postoperative videourodynamic parameters were compared. In addition, BCI and VE were also calculated from the recorded parameters.

The patients received at least one injection of no less than 200 U of BoNT-A (Allergan, Irvine, CA, USA) in the detrusor muscle at baseline under light intravenous general anesthesia in the operating room. The BoNT-A (200 or 300 U) was diluted with 30 mL of normal saline, and the diluted solution was injected into 30 sites in the bladder wall, excluding the bladder trigone [27]. The bladder volume for detrusor injection was controlled at 100 mL, and the injecting sites were separated by a 2 cm distance to cover the bladder wall as much as possible. The VUDS parameters after BoNT-A injection were also recorded for comparison of bladder function. AE was performed for those patients who wished to have greater improvements in bladder conditions, including less urinary incontinence, a lower degree of hydronephrosis, and fewer UTI episodes. The surgical procedure of AE was performed using a terminal ileal segment in a modified Hautmann’s procedure; it was published previously [28,29]. After AE, all patients received post-operative VUDS for evaluation of bladder function and treatment outcome.

This study was approved by the Ethics Committee of Tzu Chi General Hospital, Hualien, Taiwan (TCGH 098-53, 098-088, and 110-033-B, Approval date were 5 July 2010 and 1 February 2021). All intact written, informed consent was obtained prior to recruitment. All methods were performed in accordance with the relevant guidelines and regulations. Statistical analysis was applied to compare the VUDS parameters before and after BoNT-A injection or AE with Student’s t-test and a paired t-test for longitudinal comparison, using SPSS software (v. 10; SPSS, Chicago, IL, USA).

## Figures and Tables

**Table 1 toxins-14-00047-t001:** The patient distribution at different spinal cord injury levels who received BoNT-A injection or augmentation enterocystoplasty.

Level of Spinal Cord Injury	Total	Cervical SCI	Thoracic SCI	Lumbar SCI	Sacral SCI
BoNT-A to BoNT-A	103 (67/36)	43 (33/10)	55 (31/24)	5 (3/2)	0
BoNT-A to AE	25 (15/10)	5 (4/1)	15 (9/6)	3 (2/1)	2 (0/2)
Total	128 (82/46)	48 (37/11)	70 (40/30)	8 (5/3)	2 (0/2)

M/F: male/female, SCI: spinal cord injury, BoNT-A: botulinum toxin A, AE: augmentation enterocystoplasty.

**Table 2 toxins-14-00047-t002:** VUDS parameters measured before and after BoNT-A detrusor injections in patients with SCI.

Urodynamic Parameters		Total Patients with BoNT-A Injections (*n* = 128)			Male Patients with BoNT-A Injections (*n* = 82)			Female Patients with BoNT-A Injections (*n* = 46)		
		Before Injections	After Injections	*p*	Before Injections	After Injections	*p*	Before Injections	After Injections	*p*
Sensation	FSF (mL)	114 ± 66	168 ± 109	<0.001	118 ± 68	167 ± 111	<0.001	108 ± 64	169 ± 108	0.001
	FS (mL)	156 ± 91	232 ± 133	<0.001	160 ± 96	231 ± 137	<0.001	147 ± 83	232 ± 128	<0.001
	US (mL)	167 ± 97	252 ± 142	<0.001	173 ± 103	251 ± 148	<0.001	157 ± 85	254 ± 134	<0.001
Storage	CBC (mL)	212 ± 119	359 ± 186	<0.001	223 ± 127	369 ± 198	<0.001	195 ± 101	341 ± 163	<0.001
Voiding	Pdet.Qmax (cmH_2_O)	34.2 ± 26.8	22.0 ± 19.7	<0.001	34.8 ± 29.6	22.7 ± 19.7	0.001	33.1 ± 21.3	20.0 ± 20.0	<0.001
	Qmax (mL/s)	4.30 ± 5.47	3.29 ± 5.13	0.028	4.20 ± 4.11	3.24 ± 4.68	0.100	4.50 ± 7.34	3.37 ± 5.90	0.147
	PVR (mL)	156 ± 114	303 ± 198	<0.001	156 ± 115	307 ± 209	<0.001	156 ± 113	296 ± 179	<0.001
	VE	0.30 ± 0.33	0.18 ± 0.28	<0.001	0.33 ± 0.33	0.21 ± 0.29	0.001	0.25 ± 0.33	0.14 ± 0.26	0.015
Contractility	BCI	-	-	-	55.8 ± 39.1	38.9 ± 31.9	0.001	-	-	-

FSF = First sensation of bladder filling; FS = First desire to void; US = Urgency sensation; CBC = Cystometric bladder capacity; PdetQmax = Detrusor pressure at maximal flow rate; PVR = Post void residual urine; VE = Voiding efficiency; BCI = Bladder contractility index.

**Table 3 toxins-14-00047-t003:** The changes in urodynamic parameters in SCI patients who continued to receive detrusor BoNT-A injections (*n* = 103, including male and female patients).

Urodynamic Parameters		Before BoNT-A Injection (*n* = 103)	3–6 Months after BoNT-A Injections (*n* = 103)	*p*
Sensation	FSF (mL)	118 ± 67	178 ± 112	<0.001
	FS (mL)	162 ± 94	250 ± 135	<0.001
	US (mL)	175 ± 100	275 ± 142	<0.001
Storage	CBC (mL)	221 ± 124	398 ± 178	<0.001
Voiding	Pdet.Qmax (cmH_2_O)	34.3 ± 27.0	20.3 ± 17.3	<0.001
	Qmax (mL/s)	4.42 ± 5.82	3.17 ± 5.12	0.019
	PVR (mL)	162 ± 117	341 ± 195	<0.001
	VE	0.30 ± 0.33	0.16 ± 0.27	<0.001
Contractility	BCI ^1^	56.4 ± 42.6	36.2 ± 32.0	<0.001

FSF = First sensation of bladder filling; FS = First desire to void; US = Urgency sensation; CBC = Cystometric bladder capacity; PdetQmax = Detrusor pressure at maximal flow rate; PVR = Post void residual urine; VE = Voiding efficiency; BCI = Bladder contractility index. ^1^ BCI was applied only to male patients (*n* = 67).

**Table 4 toxins-14-00047-t004:** The urodynamic parameters in SCI patients who converted from BoNT-A injection to augmentation enterocystoplasty (*n* = 25, including male and female patients) at times before BoNT-A injection, after BoNT-A injection and before AE, and after AE.

Urodynamic Parameters		Before BoNT-A Injection (*n* = 25)	After BoNT-A Injection, Before AE (*n* = 25)	*p*	After AE (*n* = 25)	*p* *
Sensation	FSF (mL)	98 ± 59	127 ± 90	0.093	218 ± 137	0.041
	FS (mL)	130 ± 74	153 ± 95	0.293	336 ± 136	0.001
	US (mL)	135 ± 76	157 ± 99	0.325	365 ± 130	<0.001
Storage	CBC (mL)	175 ± 88	196 ± 123	0.401	484 ± 126	<0.001
Voiding	Pdet.Qmax (cmH_2_O)	33.6 ± 26.6	27.5 ± 27.2	0.347	9.6 ± 11.4	0.005
	Qmax (mL/s)	3.84 ± 3.7	3.76 ± 5.2	0.932	2.1 ± 4.8	0.048
	PVR (mL)	132 ± 99	144 ± 118	0.594	459± 147	<0.001
	VE	0.31 ± 0.34	0.27 ± 0.33	0.481	0.05 ± 0.11	0.005
Contractility	BCI ^1^	52.8 ± 35.8	46.3 ± 36.4	0.401	20.6 ± 26.5	0.004

* *p* value of comparison of the urodynamic parameters between the subgroup after BoNT-A injection before AE and the subgroup after AE. ^1^ BCI was applied only to male patients (*n* = 15) in these 25 patients.

**Table 5 toxins-14-00047-t005:** Comparison of the changes of urodynamic parameters between SCI patients who received BoNT-A—BoNT-A and BoNT-A—AE treatment.

Urodynamic Parameters		Changed UDS Value of BoNT-A to BoNT-A (*n* = 103)	Changed UDS Value of BoNT-A to AE (*n* = 25)	*p*
Sensation	FSF (mL)	60 ± 115	86 ± 169	0.527
	FS (mL)	89 ± 136	177 ± 189	0.065
	US (mL)	100 ± 139	202 ± 197	0.042
Storage	CBC (mL)	177 ± 171	288 ± 173	0.004
Voiding	Pdet.Qmax (cmH_2_O)	−14.0 ± 26.2	−17.9 ± 27.5	0.527
	Qmax (mL/s)	−1.24 ± 5.28	−2.25 ± 4.77	0.430
	PVR (mL)	179 ± 188	325 ± 153	0.001
	VE	−0.14 ± 0.31	−0.24 ± 0.34	0.156
Contractility	BCI ^1^	−20.2 ± 40.8	−27.5 ± 37.7	0.461

^1^ BCI was applied only to male patients (*n* = 15) for BoNT-A to AE and (*n* = 67) BoNT-A to BoNT-A.

**Table 6 toxins-14-00047-t006:** The videourodynamic characteristics and the changes of urodynamic parameters after BoNT-A injections between patients with chronic spinal cord injury who underwent augmentation enterocystoplasty and who continued detrusor botulinum toxin A injections.

VUDS Characteristics	BoNT-A to AE (*n* = 25)	BoNT-A to BoNT-A (*n* = 103)	*p*
Contracted and trabeculated bladder in filling phase	24/25 (96%)	17/103 (16.5%)	<0.001
Bladder compliance (<15 mL/cmH_2_O)	11/25 (44%)	25/103 (24.3%)	0.049
Vesicoureteral reflux in filling phase	10/25 (40%)	9/103 (8.7%)	<0.001
Dilated proximal urethra and tight external sphincter	23/25 (92%)	89/103 (86.4%)	0.736
Tight external sphincter	22/25 (88%)	56/103 (54.4%)	0.002
Change of FSF (mL)	28.7 ± 82.1	59.6 ± 115.0	0.207
Change of FS (mL)	22.6 ± 105.0	89.1 ± 135.5	0.024
Change of US (mL)	22.1 ± 109.9	99.7 ± 138.7	0.010
Change of CBC (mL)	20.7 ± 121.2	176.8 ± 171.0	<0.001
Change of Pdet.Qmax (cmH_2_O)	−6.08 ± 31.7	−14.0 ± 26.2	0.196
Change of Qmax (cmH_2_O)	−0.08 ± 4.66	−1.24 ± 5.28	0.314
Change of PVR (mL)	11.6 ± 107.5	179.3 ± 188.1	<0.001
Change of VE (%)	−5.12 ± 35.8	−13.6 ± 30.5	0.231
Change of BCI	−6.48 ± 37.9	−20.2 ± 40.8	0.128

FSF = First sensation of bladder filling; FS = First desire to void; US = Urgency sensation; CBC = Cystometric bladder capacity; PdetQmax = Detrusor pressure at maximal flow rate; PVR = Post void residual urine; VE = Voiding efficiency; BCI = Bladder contractility index.

## Data Availability

Data are available on request to the corresponding author.

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
