# Peer review of "Video-Urodynamic Characteristics and Predictors of Switching from Botulinum Neurotoxin a Injection to Augmentation Enterocystoplasty in Spinal Cord Injury Patients"

_toxins, 2022, doi:10.3390/toxins14010047_

Round 1
Reviewer 1 Report
ABSTRACT: the first sentence could be equivocal. BonTA injection and augmentation enterocystoplasty are alternative to pharmacotherapy but NOT to CIC, cause CIC should be continued in patients with detrusor areflexia induced by BoNTA and in those submitted to AE with incomplete voiding, as You correctly reported in the following paragraph (introduction).
You concluded in the abstract that patients should shift to AE to gain better QoL. It could be correct. However, this is not a conclusion coming from Your investigation cause You did not assess QoL. You should specify that this statement already comes from Literature.
INTRODUCTION: typical urodynamic finding (if You simultaneously record pelvic and external urethral sphincter electromyography) in SCI patients is detrusor overactivity with striated sphincter dyssynergia. You reported “incompetent and relaxing urethra”. Therefore, You probably should specify, according to Dorsher (that You cited as reference), that mixed neurogenic bladders type B are those associated to a flaccid urinary sphincter.
Considering that Your cohort is made by males and females, You should refer not to Guidelines dedicated to women, but to International Guidelines including management also of men with Neurogenic Bladder (ICI).
RESULTS: I think that the description of the population (at least in my opinion) is confounding.
Probably You should add one table to clarify how patients are distributed. Protocol is not standardized considering that patients have been not assessed by means of VUDS after BoNTA at the same time interval, and considering that patients differ about number of BoNTA injections received.
What do You mean by “voiding phase parameters” ?
Patients affected by SCI cannot voluntarily induce micturition. Therefore, detrusorial pressure at bladder neck opening and at maximum flow that You recorded by VUDS are the expression of NDO inducing loss of urine. Patients submitted to BoNTA and continuing to suffer from NDO and UI are not adequately cured.
You reported in the discussion that Authors suggested that increasing storage pressure after BoNTA injection should be considered to switch into AE. However I did not find in Your database the values of the intravesical pressure recorded during bladder filling and its relationship with the bladder volume (compliance). Another parameter that You did not take into consideration is the maximum amplitude of involuntary detrusor contractions of patients with persisting NDO. Don’t You consider these parameters of importance to assess bladder dysfunction in SCI individuals ?
Considering detrusor / sphincter dyssynergia, have all the patients of this cohort been submitted to electromyography during VUDS ? What do you think about the chance of infiltrate the urethral sphincter by BoNTA in those who suffer from high intravesical bladder pressures secondary to severe dyssynergia as assessed by electromyography and radiologic observation ?
A very important aspect of therapy, especially in SCI patients with bladder disorders, is quality of life. In this study there is a lack of data coming from symptomatic questionnaires and QoL scores. Do You consider that this aspect could have somehow influenced or limited the value of Your investigation ? Please comment.
DISCUSSION: in this study you report data from VUDS. VUDS have specific indications and are characterized by higher costs than UDS. Then, which is the real advantage that VUDS add to SCI patients evaluation respect to a simple urodynamic study ? In the discussion and also in the conclusion You did not report any morhpologic outcome as a contributing factor for switching patients from BoNTA to AE. Therefore do You still consider VUDS highly recommended or not in the algoritm for choosing which patients need to switch from BoNTA to AE ?
CONCLUSION: You reported a lot of data from a large cohort of patients. Don’t You consider that the final message after all this risk to appear too much simple ? I think that Literature provides already enough data that clearly show the need to shift patients to another treatment when urodynamic data after BoNTA are not safe.
I’m sorry for these comments. Some indications could appear provocative but I really want to enhance the value of this manuscript and avoid to keep in it wrong concepts. Please make easier the understanding of the procotol, the classification of polulation subgroups, do not confound the readers with so many data without a conclusive message. Remember that this is a retrospective analysis and for this reason is already of limited power.
Reviewer 2 Report
In this study the authors analysed the influence of various video-urodynamic study (VUDS) parameters in spinal cord injury (SCI) patients who continuously received repeat BoNT-A detrusor injections or switched to AE to improve their bladder conditions. They compared the changes in the urodynamic parameters before and after each mode of treatment. SCI patients receiving BoNT-A injections but experiencing little improvements in their urodynamic parameters should consider switching to AE to achieve a better quality of life.
This study was very substantial improved, but still has some limitations and needs some improvements before publishing:
Introduction
Lines 36-37: please add multiple references at the end of each these sentences.
Materials and Methods
Lines 303-304: you said: This study was approved by the Ethics Committee of Tzu Chi General Hospital, Hualien, Taiwan (TCGH 098-53, 098-088, and 110-033-B). Please add also to this sentence the exactly date of the permission of all your experiments.
Results
Please say “Table” in the whole manuscript.
Discussion
Lines 182-184: please correct this sentence.
Lines 194-198: please say “200 U”.
Round 2
Reviewer 1 Report
INTRODUCTION:
- I think the verb "render" needs to be followed by an "attribute" OR a "direct object". To render what ? Please improve. Anyway the sentence has no significance.
- Aiming to a better comprehension and to simplify the goal of this study for the readers, I would like at the end of the introduction to change the last sentence, something similar to "Basing on pre and post operative videourodynamic investigation, aim of the present study was to assess if SCI patients submitted to BoNTA achieve better functional outcomes from Bo-NTA re-injection or from augmentation enterocystoplasty".
RESULTS:
I think that also this part needs to be simplified.
Something very simple to be understood. For example:
"In this study a cohort of 128 SCI patients has been recruited. Clinical (lesional level) and demographic characteristics (male, age, sex) of the patients are reported in Table 1. All these patients have been submitted to VUDS with external sphincter elctromyography before and after Bo-NTA injections. The following VUDS parameters have been assessed: badder compliance, maximum cystometric capacity, detrusor overcativity (maximum detrusorial pressure during involuntary detrusor contractions), morphologic characteristics of the bladder, evidence or not of vesicoureteral reflux, morphologic aspect of the bladder and of the urethra during leakage, bladde residual volume. Pre and postoperative videourodynamic parameters have been compared. Of these 128 patients, 25 moved from Bo-NTA injection to AE because a decreased bladder capacity and significant bladder compliance reduction was observed."
If you should be more clear in the description of the protocol and refer the reader to the Tables to describe population and outcomes I think is better.
I think that some details (i.e. the neurologic lesional level, the rate of males and females, the number of Bo-NTA treatments could be reported in the Tables. Readers need to easily understand the protocol and the aim of the study.
If You want to be detailed on all these parameters you should add all in the "methods" paragraph that should be inserted before the RESULTS.
PAY ATTENTION: in the legend of Table 1 you wronlgy reported "ord" instead of the correct word: "cord" (is written without the c).
I also suggest You to not report the data of the so called "voiding phase". I already explained this: SCI patients cannot induce voluntarily the micturition. Therefore is no need to differentiate detrusorial pressures reached during DO from those of the "voiding phase" because in almost the total amount of the SCI the "voiding phase" is the consequence of DO and therefore it is not a "micturition" but a "leakage of urine". In these patients, Cystometric parameters, electromyographic outcomes and (in case of videourodynamic studies) morphological observations are enough to describe the functional pattern.
I also suggest You to be less detailed in the description of VUDS results. You should express the concept in the paragraph and let the reader to refer to the Tables to see the data (numbers). If you report the numeric data either in the text AND also in the tables might be confounding.
Please use the word "rate" instead of "percentage"
I suggest You to remove the picture of the videourodynamic test from the manuscript. The image clearly shows that the transducers have not been correctly balanced (zero atmosphere - please refer to www.ics.org).
However, You could keep in the paper the images, better if magnificated.
DISCUSSION
After the second sentence You should insert the "dot".
You should not write "This present study". Please choose from "This study" or "The present study".
In the sentence "Consequently in this study....", please change "urodynamic" with "videourodynamic" (row 223). The same at row 284.
CONCLUSION:
Row 298 put "of" instead of "in".
Change accordingly to Therefore or then. If You use accordingly you should specify Accordingly to .. something (row 302).
Change "near future" to "next future".
Please simplify the text. Remove data that can be resumed by tables. Do not confound the reader. It is important to enhance the concepts and messages you want to explain and comunicate without not necessary details in the text.
I think that a language review is still useful.
I hope to help You to achieve a satisfying result.
Reviewer 2 Report
Thank you for your corrections.
Author Response
Thank you for reviewer's comment.
This manuscript is a resubmission of an earlier submission. The following is a list of the peer review reports and author responses from that submission.
Round 1
Reviewer 1 Report
Main limit of this type of study is the retrospective design.
Furthermore, population is not homogeneous (different NDO conditions) and consequently the subgroups are small.
Evaluation protocol is also not homogeneous cause not all patients have been submitted to the same controls at the same time.
Sometimes the repor of results appears not clear and citations are not appropriate (i.e. when you refer to manuscripts regarding OAB and not NDO in the Discussion.
Sometimes also the language seems to not appropriate (bladder filling phase parameters are better expressed as “cystometric parameters”.
I think that the content of the manuscript is poor to support a publication with a strong scientific power.
Author Response
Point 1: Main limit of this type of study is the retrospective design.
Response 1: We agree with the reviewer’s comment that a retrospective cohort study inherits biases and confounders comparing to a prospective study. In this retrospective study, we tried to reflect the evaluation and treatment outcome after serial botulinum toxin A (BoNT-A) injections by observation of the video-urodynamic study of patients with refractory neurogenic detrusor overactivity. Hopefully in the near future, prospective study related to predict timing of switching BoNT-A injections to augmentation enterocystoplasty will take place, in order to depict a whole picture of the disease entity.
Point 2: Furthermore, population is not homogeneous (different NDO conditions) and consequently the subgroups are small. Evaluation protocol is also not homogeneous cause not all patients have been submitted to the same controls at the same time.
Response 2: We appreciate the reviewer’s important point. We agree that it is difficult to control the patient numbers under retrospective recruitment. Nevertheless, we tried to recruited all patients with spinal cord injury, instead of other neurological diseases. In real world setting, those patients who would like to receive augmentation enterocystoplasty (AE) are fewer than only BoNT-A injections, which made the patient’s number of subgroups of receiving AE small. The evaluation protocol in this research was performing video-urodynamic study before and after BoNT-A injections or AE. All patients receiving BoNT-A injections were recruited and the follow-up duration is till the year of 2020.
Point 3: Sometimes the report of results appears not clear and citations are not appropriate (i.e. when you refer to manuscripts regarding OAB and not NDO in the Discussion.
Response 3: Thank you for the reviewer’s reminding. We revise the citation of the Nitti et al [7] into different citation regarding the NDO, which would be more appropriate for this manuscript as mentioned by reviewer. (Revised manuscript, Line155, Line 159-161)
Point 4: Sometimes also the language seems to not appropriate (bladder filling phase parameters are better expressed as “cystometric parameters”.
Response 4: We appreciate the reviewer’s suggestion. In order to revise the wording appropriately, we followed the wording of urodynamic study with reference to the published article by Drake et al. (Fundamentals of urodynamic practice, based on International Continence Society good urodynamic practices recommendations. Neurourology and Urodynamics. 2018;37:S50–S60. ) We revised “The VUDS parameters of filling phase” into “The VUDS parameters of sensation” (Revised manuscript, Line 96, Line 171, Line 209)
Point 5: I think that the content of the manuscript is poor to support a publication with a strong scientific power.
Response 5: Thank you for the critical comment. The aim of this study is to identify which spinal cord injured (SCI) patients with NDO should be switched to undergo augmentation enterocystoplasty (AE) in order to improve the bladder capacity and quality of life. Patients who switched from detrusor BoNT-A injections to AE had greater improvements in their urodynamic parameters when compared with those who continued with repeat BoNT-A injections. Accordingly, SCI patients receiving BoNT-A injections but experiencing little improvements in their urodynamic parameters should consider switching to AE to achieve a better quality of life. I think this is important for urologist to select appropriate SCI patients and perform AE at the appropriate timing. The results of this study are scientifically sound.

Reviewer 2 Report
In this study, the authors compared the changes in the urodynamic parameters before and after BoNT-A detrusor injections or augmentation enterocystoplasty in patients with spinal cord injury. The subject of this study is quite important; patients with spinal cord injury have severe bladder issues and they are dependent on assisted bladder emptying.
The main conclusion of this manuscript is that patients receiving BoNT-A injections but having little improvements in their urodynamic parameters should switch to augmentation enterocystoplasty to improve their bladder conditions. This study suggest recommendations that could have a positive impact on bladder related function and help the patients to achieve a better quality of life.
After careful reading of this manuscript, my opinion is that this study is valuable and well written, therefore deserving to be published.
Author Response
We are happy to hear that and grateful for your appreciate.
Reviewer 3 Report
Elegant and nice study, relevant for the field.
Author Response
We are grateful for your appreciate and acceptance.
Reviewer 4 Report
Botulinum neurotoxin A (BoNT-A) injection and augmentation enterocystoplasty (AE) are alternative and effective management strategies for neurogenic detrusor overactivity (NDO) refractory to clean intermittent catheterization with pharmacotherapy. Most of patients with spinal cord injury (SCI) may, however, prefer BoNT-A injections to AE, due to its less invasive characteristics. In the present study the authors evaluated the influence of various video-urodynamic study (VUDS) parameters in SCI patients who continuously received repeat BoNT-A detrusor injections or switched to AE to improve their bladder conditions. Further, they compared the changes in the urodynamic parameters before and after each mode of treatment.
The results of the study are new. Nevertheless, this manuscript needs substantial improvements and corrections before publishing may be possible.
General points:
Please do your References list at the end of the manuscript according to “Toxins”.
Please add to your manuscript List of Abbreviations.
Please say in the whole manuscript including also the title of the manuscript: Botulinum neurotoxin A (BoNT-A).
Special points:
Keywords: please add also to keywords: patients; botulinum neurotoxin A; urodynamic parameters
Introduction
Lines 32-41: please add multiple references at the end of each these sentences.
Lines 47-49: please add multiple references at the end of this sentence.
Lines 50-51: please add multiple references at the end of this sentence.
Lines 58-63: please add multiple references at the end of each these sentences.
Lines 52-56: please describe exactly all these studies.
Lines 66-68: please describe exactly all these studies.
Results
Lines 74-76: please describe very exactly all experimental groups with exactly patient’s numbers and the numbers of females and males in each.
Table 1, Table 2 and Table 3: please demonstrate your results in this table separately for female and male patients.
Lines 106-108: please add also the significant values to this result.
Conclusions
Please add also the future perspectives to this section.
Please add also the meaning of your results for clinicians.
Materials and Methods
Lines 240-242: please add the exactly information about the BoNT-A injections numbers and the distance between the each injection for each patient participated in your study.
Lines 240-242: please add multiple references at the end of this sentence.
Lines 253-254: please add multiple references at the end of this sentence.
Lines 251-252: please
Author Response
Point 1: Please do your References list at the end of the manuscript according to “Toxins”..
Response 1: Thank you for the reviewer’s reminding. As following the instruction for authors, we carefully revised and re-checked the reference list. (Revised version, line 313-386)
Point 2: Please add to your manuscript List of Abbreviations.
Response 2: We appreciate the reviewer’s comment and added the List of Abbreviations into the Appendix A section. (Revised version, line 290-309)
Point 3: Please say in the whole manuscript including also the title of the manuscript: Botulinum neurotoxin A (BoNT-A).
Response 3: Thank you for the reviewer’s reminding and we revised Botulinum toxin A into Botulinum neurotoxin A. (Revised version, Line 3, Line 5, Line 25, Line 48)
Point 4: Keywords: please add also to keywords: patients; botulinum neurotoxin A; urodynamic parameters
Response 4: Thank you for the comment, and we revised the keywords as your suggestion. (Revised version, Line 25-26)
Point 5: Lines 32-41: please add multiple references at the end of each these sentences. Lines 47-49: please add multiple references at the end of this sentence. Lines 50-51: please add multiple references at the end of this sentence. Lines 58-63: please add multiple references at the end of each these sentences.
Response 5: We appreciate the reviewer’s comment and we added the reference to each sentence as reviewer’s suggestion. (Revised version, Line 33, Line 36, Line 39, Line 42, Line 52, Line 66, Line 69, Line 71)
Point 6: Lines 52-56: please describe exactly all these studies. Lines 66-68: please describe exactly all these studies.
Response 6: Thank you for the comment, and we have further described these studies in more detail. (Revised version, Line 52-62, Line 74)
Point 7: Lines 74-76: please describe very exactly all experimental groups with exactly patient’s numbers and the numbers of females and males in each.
Response 7: Thank you for the comment, and we have further described the experiment groups in detail with gender distribution. (Revised version, Line 80-85)
Point 8: Table 1, Table 2 and Table 3: please demonstrate your results in this table separately for female and male patients.
Response 8: Thank you for the comment, we have demonstrated the data by different genders in table 1, 2, and 3, accordingly. (Revised version, table 1, table 2, table 3)
Point 9: Lines 106-108: please add also the significant values to this result.
Response 9: Thank you for the comment, and we have added the significant p value in the description. (Revised version, Line 119)
Point 10: Please add also the future perspectives to this section. Please add also the meaning of your results for clinicians.
Response 10: We are grateful for the suggestion, and we have added the meaning of our results for clinicians and the future perspectives to the conclusion section. (Revised version, Line 233-244)
Point 11: Lines 240-242: please add the exactly information about the BoNT-A injections numbers and the distance between the each injection for each patient participated in your study.
Response 11: Thank you for the comment, The BoNT-A (200 or 300 U) was diluted with 30 mL of normal saline and the diluted solution was injected into 30 sites in the bladder wall, excluding the bladder trigone. The bladder volume for detrusor injection was controlled at 100ml, and the injecting sites were separated by 2 cm distance to cover the bladder wall as much as possible. (Revised version, Lines 259-265)
Point 12: Lines 240-242: please add multiple references at the end of this sentence. Lines 251-252: please add multiple references at the end of this sentence.
Response 12: We appreciate the reviewer’s comment and we added the reference to each sentence as reviewer’s suggestion. (Revised version, Line 263, Line 281-284)
Round 2
Reviewer 1 Report
I think the goal of this manuscript is interesting but the design of the study, the protocol, the retrospective plan, the mixed population still not convince me.
Neurogenic bladder management is not so difficult. In spinal cord injured patients with suprasacral lesions the aim of the treatment is to maintain a low bladder pressure (possibly below 40 cmH2O), reduce hyperactivity, prevent vesico-renal reflux, and in patients with DSD reduce post micturition residual of urine. Furthermore, in parients with significant bladder residue self catheterization is recommended.
To achieve these goals guidelines are already clear: after conservative treatment, antimuscarinics are mandatory followed by botulinum and AE. The decision making among these treatments is easy: when urodyamic parameters do not satisfy safety for the urinary system. Therefore, if clinical signs and symptoms or urodynamic parameters do not improve during treatment, a new approach is indicated following international guidelines.
This study would have provided some further suggestions but I think it risks to be confounding. Results of the voiding phase of the patients are reported. However, it is expected that many of these patients do not present micturition after botulinum toxin treatment (urinary retention induced by the treatment). There is a lack of informations regarding morpho- functional parameters taken into consideration in the decision making process (the study is based on Videourodynamics).
Reviewer 4 Report
Thank you for corrctions.